# The Role of BIA Analysis in Osteoporosis Risk Development: Hierarchical Clustering Approach

**DOI:** 10.3390/diagnostics13132292

**Published:** 2023-07-06

**Authors:** Giacinto Angelo Sgarro, Luca Grilli, Anna Antonia Valenzano, Fiorenzo Moscatelli, Domenico Monacis, Giusi Toto, Antonella De Maria, Giovanni Messina, Rita Polito

**Affiliations:** 1Department of Economics, Management and Territory (DEMeT) and Grant Office, University of Foggia, 71121 Foggia, Italy; giacinto.sgarro@unifg.it; 2Department of Clinical and Experimental Medicine, University of Foggia, 71122 Foggia, Italy; anna.valenzano@unifg.it (A.A.V.); fiorenzo400@gmail.com (F.M.); rita.polito@unifg.it (R.P.); 3Department of Humanities, Letters, Cultural Heritage, Educational Sciences, University of Foggia, 71100 Foggia, Italy; domenico.monacis@unifg.it (D.M.); giusi.toto@unifg.it (G.T.); 4Section of Human Physiology and Unit of Dietetics and Sports Medicine, Department of Experimental Medicine, University of Campania “Luigi Vanvitelli”, 80131 Naples, Italy; antodemar82@gmail.com

**Keywords:** clustering, classification, body composition (BC), bioelectrical impedance analysis (BIA), osteoporosis

## Abstract

Osteoporosis is a common musculoskeletal disorder among the elderly and a chronic condition which, like many other chronic conditions, requires long-term clinical management. It is caused by many factors, including lifestyle and obesity. Bioelectrical impedance analysis (BIA) is a method to estimate body composition based on a weak electric current flow through the body. The measured voltage is used to calculate body bioelectrical impedance, divided into resistance and reactance, which can be used to estimate body parameters such as total body water (TBW), fat-free mass (FFM), fat mass (FM), and muscle mass (MM). This study aims to find the tendency of osteoporosis in obese subjects, presenting a method based on hierarchical clustering, which, using BIA parameters, can group patients who show homogeneous characteristics. Grouping similar patients into clusters can be helpful in the field of medicine to identify disorders, pathologies, or more generally, characteristics of significant importance. Another added value of the clustering process is the possibility to define cluster prototypes, i.e., imaginary patients who represent models of “states”, which can be used together with clustering results to identify subjects with similar characteristics in a classification context. The results show that hierarchical clustering is a method that can be used to provide the detection of states and, consequently, supply a more personalized medicine approach. In addition, this method allowed us to elect BIA as a potential prognostic and diagnostic instrument in osteoporosis risk development.

## 1. Introduction

Osteoporosis is a musculoskeletal disorder characterized by low bone mass and microarchitectural deterioration of bone tissue. It is a pathology associated with aging, it is much more frequent among females than males, and it is associated with inflammatory disorders also induced by an unhealthy lifestyle, obesity and overweight. It is a global health problem associated with aging and is three times more common in women than in men [1]. The prevalence of osteoporosis is expected to increase as the population ages. Osteoporosis patients with fragility fractures and vertebral deformities have impaired activity of daily living and quality of life (QOL) [2,3]. In the literature, it was reported that the incidence of musculoskeletal disorders has increased by 17.7% in the last ten years [4]. In addition, in 2010, the total number of individuals ≥ 50 years of age, the age group at highest risk for osteoporotic fractures, reached 158 million. This number is expected to double by 2040 [4,5]. Therefore, early screening and intervention for osteoporosis have become important clinical tactics to keep fracture rates and related morbidity as low as possible in this population. In this scenario, a parameter such as bone mineral density (BMD), defined as the mass of bone mineral per unit volume, plays an important role. Dual-energy X-ray absorptiometry (DEXA) is a commonly used method for measuring bone mineral density (BMD) and monitoring bone health. DEXA scans are widely used for several purposes related to bone health. They are commonly used for diagnosing osteoporosis, a condition characterized by low BMD and an increased risk of fractures. DEXA can detect bone loss and provide valuable information about an individual’s fracture risk. It is considered the gold standard for BMD measurement due to its precision and accuracy. This is the gold standard, indicating skeletal metabolic status and being used to analyze change in bone mass over time [4,5,6].

This parameter is associated with different factors such as age, weight, diet, exposure to sunlight, early menopause, smoking, alcohol, genetic factors, gender and exercise [6]. Among these factors, heredity, gender, and age are immutable, while weight, diet, exercise, exposure to sunlight and lifestyle are modifiable. Body composition (BC) indicators such as body mass index (BMI), fat-free mass (FFM), fat mass (FM), and skeletal muscle mass index (SMI) are the result of the combined effects of non-modifiable and modifiable factors on the human body [7,8,9,10]. Obesity is a long-term multifactorial chronic disease characterized by an energy imbalance due to an excess of caloric intake compared with energy expenditure and the deregulation of other metabolic parameters, such as altered lipid profile, increased insulin resistance, and chronic proinflammatory state [11,12,13]. There are two different types of obesity: central (abdominal adipose tissue rather than appendicular accumulation), and visceral. Subcutaneous and visceral adipose tissue compartments comprise the central adipose tissue that results from this process. Because visceral adipose tissue (VAT) may result in a subclinical condition of systemic inflammation, the severity of obesity-related disorders is directly correlated with body composition [14,15]. Furthermore, the analyses of body composition (BC) are fundamental in nutritional status evaluation both in physiological and pathophysiological conditions. BC is a complex of unmodifiable factors such as heredity, sex, aging, and of acquired lifestyle factors which are very modifiable [16]. In fact, a range of weight-control strategies, notably exercise, can change BMI and SMI [17]. Thus, BC can be viewed as an aggregate outcome of the cumulative effect of unmodifiable and modifiable factors in the human body. Therefore, BC indicators may be useful not only as early predictors of BMD risk, but also as indicators of BMD intervention effectiveness. The interrelationship of body composition and osteoporosis is complex and multifactorial, possibly because of differences in ethnicity, nutrition, lifestyle habits, and obesity [17]. The most used methods to evaluate BC in clinical practice are based on bicompartmental models and measure, directly or indirectly, FM and FFM. Bioelectrical impedance analysis (BIA) is one of the methodologies that can be used to calculate BC parameters. BIA is a non-invasive technique that measures human body bioelectrical conductivity value, and it provides an accurate estimate of BC [18]. Detecting groups (clusters) of closely related objects is an important problem in bioinformatics and data mining in general [19]. Clustering is a class of methods used to identify homogeneous groups in a set of unlabeled elements [20]. Mathematically, the elements of a set can be seen as points in a multidimensional space that a clustering technique attempts to group in such a way that points in a single group have a natural relation to one another, and points in other groups are somehow different [21]. Similarity or dissimilarity among elements is measured using distance metrics. Generally, clustering is effective when both the variability among the same cluster elements (intra-cluster variability) is as low as possible, and the variability among different clusters (inter-cluster variability) is as high as possible [6].

In the literature, clustering techniques have been used successfully as a means of pattern recognition in many engineering, medical and biological fields, such as the identification of muscle activation patterns or the detection of biologically meaningful gene clusters in genomic data from several species such as yeast and mouse [21,22,23,24]. However, minimal clustering methods are employed to identify BIA data patterns [25]. Hierarchical clustering belongs to the class of cluster analysis algorithms, and it is a clustering approach that aims to build a hierarchy of clusters by joining or dividing clusters at each iteration [26]. In agglomerative clustering, at the beginning, each element starts from a separate cluster, and for each iteration, a pair of clusters is merged. On the contrary, in divisive clustering, all elements start by belonging to one cluster at the beginning, and for each iteration, a cluster is split in two [26]. A metric and a linkage criterion are needed to combine or split clusters. A metric represents the distance between pairs of elements [27], while a linkage criterion represents clusters’ dissimilarity as a function of the elements’ pairwise distances between different clusters [28]. A possible graphical output of a hierarchical clustering algorithm is the dendrogram [23], a tree diagram representing the arrangement of the clusters obtained from an agglomerative hierarchical clustering method based on a proximity measure [29]. In summary, it can be said that hierarchical clustering allocates elements into a dendrogram whose branches are the clusters, and cutting the tree at a certain height means choosing the iteration connected to a certain number of clusters [21]. Given the strong correlation between body composition, obesity, osteoporosis, and vitamin D, supported by many scientific research [5,6,7,8,17], the aim of this study is to find the tendency of osteoporosis in obese subjects, presenting a method based on agglomerative hierarchical clustering which, using BIA parameters, can group patients who show homogeneous characteristics in order to individuate potential biomarkers of disease, using BIA as a prognostic and diagnostic instrument.

## 2. Materials and Methods

### 2.1. Participants

This study involved the recruitment of 46 individuals who were obese (BMI > 30), with ages ranging from 23 to 62 years. Among the participants, there were 14 males and 32 females. The study took place at the Laboratory of Physiology, Department of Clinical and Experimental Medicine at University of Foggia. The local Institutional Ethics Committee approved the study (Azienda Ospedaliera-Universitaria “Ospedali Riuniti”, Foggia, Independent Ethics Committee; protocol number that was attributed by the ethics committee: no. 440/DS). All subjects recruited for the investigation provided both written and oral information regarding the possible risks and discomforts and were ensured that they were free to withdraw from the study at any time.

### 2.2. Exclusion Criteria

Participants were excluded if they had a prior medical history of renal insufficiency, hyperuricemia, severe hepatic insufficiency, type 1 or 2 diabetes mellitus treated with insulin, atrioventricular block, heart failure, cardiovascular and cerebrovascular diseases, unbalanced hypokalemia, hypo- or hyperthyroidism, chronic treatment with corticosteroid drugs, severe mental disorders, neoplasms, pregnancy, or if they were lactating.

### 2.3. Anthropometric Measurements

The height, weight, BMI, and waist circumference of the 46 obese participants were recorded. Body weight was measured in a fasting state in the morning with a mechanical balance (±0.1 kg, SECA 700, Hamburg, Germany). The subjects were laid down on the couch, and we waited 1 min before starting the procedure, to allow the liquids to stabilize in the supine position. It was recommended not to drink large quantities of water before the test and to empty the bladder. Additionally, subjects were advised not to drink coffee or alcohol before the test and not to engage in physical activity before the test.

Participants were positioned lying face-up with their arms positioned alongside their bodies, ensuring they did not contact the body. Their legs were extended and slightly separated. To ensure precise application, the electrodes were carefully placed on the back of the hand and foot after thoroughly cleaning the skin.

The BMI was calculated as body weight divided by height squared (kg/m^2^), with categories in accordance with the World Health Organization guidelines, and results ranged from 23.6 to 39.3 kg/m^2^. To define participants with alterations in BMI and circumferences, the reference intervals normalized to the age of each participant were applied [14].

### 2.4. Analysis of Body Composition

Body composition parameters (body fat mass, body lean mass, and bone mineral content) were calculated using a body fat caliper and using bioelectrical impedance analysis (Quantum V Segmental Bia). In addition, as previously reported, we also performed DEXA and vitamin D serum levels evaluation [15].

### 2.5. Hierarchical Clustering Algorithm: Clustering for Prototype Patients’ Recognition

We present a method for grouping patients with similar health condition and extracting prototypes (Principal states) based on the application of a hierarchical clustering algorithm. Considering a specific patient, the algorithm consists of four phases: (a) dataset preparation, (b) dataset clustering, (c) tree cutting, (d) Patient prototypes (Principal states) extraction (Figure 1).

#### 2.5.1. Dataset Preparation

All anthropometric and BIA patients’ data are pooled together. The available features were: gender, age, height, weight, BMI, TBW, FFM, FM, MM and BMI, TBW, FFM, FM, and MM percentage. We decided to select eight variables to be used by the algorithm: age, height, weight, and TBW, FFM, FM, and MM percentage. This is because we did not want gender affections. BMI information was in height and weight, and percentage values were more appropriate than kilograms for different subjects. Dataset was normalized using the following Equation (1):(1)$$x_{i,norm}=\frac{\left(x_{i}-x_{min}\right)}{\left(x_{max}-x_{min}\right)}$$
where *x_i_* is the value of feature *i*, and *x_min_* and *x_max_* are the minimum and maximum values of the feature, respectively. Every patient *j* of the dataset is identified by an array *p_j_* characterized by *m* = 8 features:(2)$$p_{j}=\left\{age,height,weight,TBW,FFM,FM,MM\right\}$$
with *j* = 1…*n*, where *n* is the total number of patients belonging to the dataset.

#### 2.5.2. Dataset Clustering

We applied agglomerative hierarchical clustering to the dataset. In the beginning, each patient is considered as a separate cluster, and, after each iteration, the two closest clusters are merged using a specific metric and a linkage criterion [6]. The metrics and linkage criteria used in this work are specified in Table 1 and Table 2. Patient *A* (p_a_) and patient *B* (p_b_) are generical subjects enclosed in distinct clusters *A* and *B*, and they can be seen as two *m*-dimensional points in an *m*-dimensional space where each dimension is a feature chosen. Using a linkage criterion, it is possible to define the distance between two clusters, *A* and *B*, as a metric-based function between the elements of the two different clusters.

To explain how this algorithm works, let us suppose we have three clusters *A*, *B* and *C* at *x*th-iteration. Cluster *A* has two elements, *p*_1_ and *p*_2_, cluster *B* has three elements, *p*_3_, *p*_4_ and *p*_5_, and cluster *C* has one only element *p*_6_. During the iteration, one of the cluster combinations (*A* − *B*, *A* − *C*, *B* − *C*) must be chosen for the merging. Using, for example, “Euclidean” metric and “single” linkage criterion, for the *A* − *B* combination, the following Euclidean distances are computed: *d*(*p*_1_, *p*_3_), *d*(*p*_1_, *p*_4_), *d*(*p*_1_, *p*_5_), *d*(*p*_2_, *p*_3_), *d*(*p*_2_, *p*_4_), *d*(*p*_2_, *p*_5_); and using the Single-linkage method, the minimum distance is considered. At this step, each cluster combination is represented by one number, and we join the couple of clusters with the minimum value.

An agglomerative hierarchical clustering algorithm, at each iteration, starting from clusters with only one element, merges together couples of clusters until one single cluster is formed: (Figure 2). The software we used for clustering is MATLAB (version R2022a), and we used the function linkage.

#### 2.5.3. Dendrogram Analysis and Cutting

For dendrogram cutting, i.e., choosing the number of clusters during an agglomerative clustering process, we developed an automatic rule for general use in a clustering process. We investigate if mathematically significant clusterizations result in biologically significant clusterization. The automatic rule is based on the idea that the tree must be cut where clusters have small intra-cluster variability, and, at the same time, there is a great distance from one another. This rule worked in the following way: at each iteration, (A) compute intra- and inter-cluster variability (Equations (3) and (4)), (B) consider maximum intra-cluster and minimal inter-cluster variabilities (Equations (5) and (6)), (C) calculate an *R_k_* index (Equation (7)) and (D) in the end, cut the tree where *R_k_* is maximum.


Automatic Rule:



(A)Compute intra- and inter-cluster variability


Intra-cluster variability is calculated as the sum of Euclidean distances of each element of the cluster from its centroid divided by the cluster cardinality:(3)$${Var}_{intra,A}=\frac{\sum_{j=1}^{\left|A\right|}\sqrt{\sum_{i=1}^{m}{\left(p_{j,i}-C_{A,i}\right)}^{2}}}{\left|A\right|}\;\;$$
where *C_A_* is the centroid of cluster *A*, defined as an *m*-dimensional vector containing the mean of the patients features inside the cluster *A*.

Inter-cluster variability between two clusters is calculated as Euclidean distance between the centroids:(4)$${Var}_{inter,A,B}=\sqrt{\sum_{i=1}^{m}{\left(C_{A,i}-C_{B,i}\right)}^{2}}\;$$


(B)Maximum intra-cluster and minimal inter-cluster variabilities are defined as follows:


(5)$${Var}_{intra,MAX}=\max\left({Var}_{intra,\;\;A}\right)$$(6)$${Var}_{inter,min}=\min\left({Var}_{intra,A,B}\right)$$
where *A* and *B* are two generic clusters at the *k*-iteration (Figure 3).


(C)$R_{k}$ index represents the ratio between the minimal inter-cluster variability and the maximum intra-cluster variability obtained during one iteration in which two clusters are merged. It is computed as follows:



(7)
$$R_{k}=\frac{{Var}_{inter,min}}{{Var}_{intra,MAX}}\;$$


We noticed that the *R_k_* values were higher in the final iterations, so we computed *R_k_* only for the last 10 iterations, as in Figure 4.

(D)Tree cutting: it is the operation of cutting the dendrogram at the point corresponding to the optimal clusterization, i.e., the number of cluster where the *R_k_* value is maximum (Figure 5). Using various combinations of distance-linkage criteria, we obtained numerous potential clusterization outcomes (as shown in Table 3), each with the potential for physiological interpretation. Considering the labor-intensive nature of physiological interpretation, we have chosen to narrow down our analysis to the most frequently observed pattern of clusterization. Specifically, we will focus on the metric-linkage combinations referred to as “2 Clusters (Clustering 1)” in Table 3. We obtained at least one interesting outcome for each combination of distance-linkage criterion. In order to physiologically interpret each of these results, we will limit our analysis to the metric-linkage combinations corresponding to: 2 Clusters (Clustering 1) of Table 3. Such a result consists of the maximum corresponding *R_k_* value in a two group clusterization. As we will show in the next paragraph, two dishomogeneous clusters were found: one composed by two elements, and the other one by the remaining elements.

#### 2.5.4. Patient Prototypes (Principal States) Extraction

We decided not to discard any cluster since it could be associated with a particular “health state”. The idea behind this is that the clustering procedure can define different health state patterns, and so it may be helpful to extract clusters’ ‘representers’, such as the cluster centroid (Figure 6), so that each cluster can be associated with a particular “health state”. From Figure 7, we can extract the peculiar characteristics that discriminate the two clusters. In Cluster 1, subjects tend to be younger and more overweight. Furthermore, they even present clearly separated values in terms of TBW, FFM, FM and MM. In particular, Cluster 1 comprises the patients who have lower TBW, FFM and MM, and higher FM.

## 3. Results

### 3.1. Physiological Interpretation

A hierarchical dendrogram algorithm produced a representation that brings forth questions like these: ‘How many useful groups are in this data?’ and ‘What are the interrelationships between groups?’ [16]. A dendrogram tree contains such information, and in this application, we will try to answer the second question. Figure 6 and Figure 7 show the characteristics of each group and the differences between the two groups. In this case, there are two patients who were considered “different” from the others.

In Table 4, the principal averaged features of participants are reported. In addition, from the analysis of the values relating to body composition, it emerged that the reference sample comprises subjects having a TBW ranging from 61.3 to 38.4%, an FFM from 52.5 to 83.7%, an FM from 16.3 to 47.5%, and finally, an MM from 33.2 to 64%.

The results show that the patients of the blue group are two women, with a weight greater than the mean value of rest of the group, but above all, they have values clearly separate from the others for the other characteristics, namely, TBW, FFT, and MM. These are much lower than the others, and the values are the minimum in the dataset. On the contrary, the FM value is the maximum in the dataset. We could define these subjects as a model of obesity. The increase in FM and the concomitant reduction in the FFM and MM in these two female subjects, with an average age of 35 years, can be a predictive model for the development of osteoporosis at an early age, as the reduction in FFM and MM also represents an alteration in bone mineralization.

### 3.2. Potential Use of Clustering Results

Cluster analysis can be used not only to recognize characteristic patterns along the features that describe a sample, but it can also be used as a basic tool for classifying or for building risk index formulas. Provided below are two purely conceptual examples which show the potential of using clustering methodologies along the two fields of applications mentioned.

Generally, the term classifier refers to a kind of system which is able, given a set of elements, to divide elements into different classes which are defined as having a certain common property as a class and in such a way that all the elements in that class fulfill that classification property [30]. In other words, given a classifier which can work correctly, when provided an element, it is able to define which class it belongs to.

One of the possible ways for performing a classification is by using a fuzzy classifier. A fuzzy classifier is a classification model used in artificial intelligence, whereby the input and output variables are described by using, for each one of them, indicators called membership functions (MFs), which allow the nuances and ambiguities present in the data (both regarding input and output) to be represented [31]. In a fuzzy classifier, MFs are connected by rules (fuzzy proposition), and the involvement of an MF within a rule is not characterized by a yes/no firing but rather represented by a probability degree or a partial certainty. This characteristic, the fuzziness of the system, makes the fuzzy classifiers very useful in contexts where there is a high degree of uncertainty, such as in speech recognition, systems performance evaluation, risk analysis, and many other contexts [32].

In this context, clustering output data (either cluster elements or centroids) could be used to define both the MFs’ shape along with the variables connected to the two different states and the fuzzy rules to classify them. Observing Figure 6, it is possible to notice that the two clusters, which, in this case, can be called classes, can be considered as separable along with the variables TBW, FFM, FM, and MM. Such information can be used for creating a classifier which has as an output a variable called “risk of development of osteoporosis in early age” that can be described by two MFs called “Risk” and “No-Risk” (Figure 8). By using the two clusters/classes centroids and the first “No-Risk” values close to the class “Risk”, it is possible to define the shape of the MFs along these four risk-correlated variables (Figure 9) and subsequently connect them by two simple fuzzy rules which connect the MFs relating to each class (Table 5 and Figure 10).

Regarding the possibility of building risk index formulas, we again want to point out how the variables TBW, FFM, FM, and MM each seem to be able to behave as a discriminator for the two classes. Thanks to that, it is virtually possible to define a threshold value between the two classes, choose a certain number of “Percentage Risk Prototypes”, and develop a formula that can be used to define an index of risk, in which the higher the index value, the greater the risk of a patient developing osteoporosis at an early age. For example, we propose a toy formula which should be as intuitive as possible by showing that the higher FM is, the lower TBW, FFM and MM are, and the higher the index of ROE (risk of development osteoporosis in early age) is:(8)$$ROE=\alpha \left(1-TBW\right)+\beta \left(1-FFM\right)+\gamma FM+\delta \left(1-MM\right)$$

For determining the formula, coefficient values (*α*, *β*, *γ*, *δ*) can be used as the cluster elements, centroids, or information, as in Table 6, for example.

In such a way, solving the system of linear equations, the following formula can be obtained:(9)$$ROE=-0.6418\left(1-TBW\right)-0.1\left(1-FFM\right)-0.0383FM+0.8135\left(1-MM\right)\;\;$$

Note: Both the example fuzzy classifier and ROE index are just conceptual examples that we want to highlight to show how the potential of clusterization is high for working on practical real-world contexts and not actual proposals, and these examples also demonstrate the restricted size of the dataset used and the need for more in-depth examinations regarding the pathology identified.

## 4. Discussion

Cluster analysis-based state classification could assist specialists in decision making but is rarely applied to BIA data [24,25,33,34]. A systematic approach for defining states of health has never been used on BIA data. This contribution fills that gap, presenting a clustering-based method used to characterize health state patterns, starting from a patient set. This method is useful for a BIA data-correct interpretation in health analysis because it allows us to obtain cluster prototypes that can be used as a means of comparison to individuate different health states. Cluster prototypes and the distribution of elements can be useful to help in defining a particular state of health. Moreover, the frequency of patient clusters in a study allows researchers to know indicatively how frequent a specific pattern is. We extracted cluster prototypes that can be used as pattern indicators, in this case to identify patients with a high risk of having/developing osteoporosis. Our results showed that 2 of the 46 patients have a very low TBW, FFT, and MM, but their FM is significantly higher than the entire dataset. In addition, these obese subjects are both females, and they have an average age of 35 years. These data are most relevant and indicate that this cluster has a possible predictive osteoporosis model. Indeed, as shown by the data reported by many studies, an alteration of adipose tissue function has been related to various human metabolism disorders such as osteoporosis [34]. It has been suggested that osteopontin influences the adipogenic process in the bone marrow of obese women, potentially contributing to the development of osteoporosis. An imbalance has been described between normal adipogenesis and osteogenesis of MSCs, prevailing adipocyte differentiation over osteoblast differentiation [35,36,37]. These results may represent a useful predictive model of osteoporosis in obese subjects in early age, and it is a point of strength of this study. A recent investigation reported a strong association between low bone mineral density (BMD) and obesity. Furthermore, it demonstrated a significant positive correlation between body fat percentage (BFP) and the likelihood of suspected osteoporosis. Additionally, BMD and BFP were found to be negatively associated [38].

Given the interesting results and, above all, the FM equal to 50% of the total weight, we also confirmed these data via DEXA and via a vitamin D serum levels evaluation [15]. These results support the data obtained by the clustering BIA analysis to elect this model as a predictive model of osteoporosis at an early age, but further studies are needed, above all due to the relatively low number of samples considered. Indeed, it is well known that vitamin D has an active role in bone mineralization [15,39]. In fact, as reported by González et al., low vitamin D serum levels are strongly correlated with obesity [39]. Vitamin D is a fat-soluble molecule that might be sequestered in the excess adipose tissue of obese adults and would be slowly released during negative energy balance [39]. In addition, a potential confounder is that obesity is also linked to an unhealthy lifestyle, characterized by less physical activity, less sun exposure and, hence, lower vitamin D concentrations [40]. In addition, our results seem to confirm the important role of BIA and body composition analysis. As reported by Marra et al., BIA measures the electrical properties of body tissues and represents a useful approach for estimating body composition parameters such as TBW and FFM. In the bicompartment model, the human body is composed of FFM, which includes, under physiological conditions, the following components: bone mineral content (≈7%), extracellular water (≈29%), intracellular water (≈44%), and visceral protein (=20%). BIA estimation of body composition is based on body fluid volume measurement using a BIA resistance value; for these reasons, an alteration of these compartments can represent prognostic or diagnostic parameters in many diseases [41].

The analysis of BC is also useful to perform measurements of bone mineral content (BMC) and density to assess bone health and diagnose osteoporosis [42]. Moreover, Fighera et al. investigated the associations between bone mass, sex hormone levels, and body composition in postmenopausal women [6]. It is also reported that there is an association between body fat percentage and the long-term risk of fractures, not only in women but also in men. This evidence allow us to believe that BIA is an effective diagnostic and prognostic tool [43]. On the other hand, it is also reported that there is a close relationship between osteoporosis and coronary artery calcification, indicating osteoporosis as a predisposing factor for cardiovascular disease [44]. In this context, the role of BIA clustering results becomes important as a prognostic factor in many different diseases. Furthermore, there are several pathophysiological and clinical features that contribute to skeletal fragility in patients with type 2 diabetes. Indeed, a combination of visceral adipose tissue accumulation and hyperinsulinemia and the accumulation of advanced glycation end products (AGEs) induced micro- and macrovascular alteration, leading to bone impairments [45]. This evidence further reinforces the significance of BIA clustering and highlights its prognostic and diagnostic value in analysis. In fact, an obese subject who is clustered for osteoporosis could also have latent hyperglycemia. Given the strong relationship between frailty, osteoporosis, body composition, and type 2 diabetes, body composition analysis could therefore indirectly have a prognostic and diagnostic role not only for metabolic but also immune and inflammatory diseases [45]. Ono et al. reported that the phase angle in postmenopausal osteoporotic patients was significantly associated with quality of life, even after adjusting for age, BMI, ASMI, and BMD [1]. Peppa et al. reported that BIA is a rapid, non-invasive and useful screening tool for determining body composition adiposity and the presence of osteo-sarcopenic features in postmenopausal women [17]. BIA analysis facilitates the evaluation of fat-free mass (FFM), a parameter that shows a positive association with bone mineral content (BMC). This association is significant due to the important role of muscle in endocrine activity. Skeletal muscle, particularly when contracting, can function as an endocrine organ and secrete myokines such as IGF-1 and irisin [46]. Irisin secreted during exercise may play a role as a messenger in the muscle–fat–skeleton–brain axis, promoting energy consumption by fat cells, the differentiation of bone cells and the suppression of the maturation of osteoclasts, thus influencing bone metabolism and enhancing bone density [47]. Furthermore, as FFM is inversely correlated to FM, then FM and adipose tissue cells are the cause of sarcopenic obesity, leading also to an inflammatory chronic state [48]. Furthermore, in the extraction of “early-age osteoporosis” prototypes, what should be considered as features comprising a measure of similarity is not yet totally defined. It is likely that we should consider MM, FM, FFM and TBW percentages, age for the connection with the disease, but not sex, because it risks excluding male patients who suffer from this pathology.

## 5. Conclusions

This investigation shows a hierarchical clustering procedure which groups subjects with homogeneous health state patterns. A quantitative evaluation of alimentary disorders and nutritional heath state patterns can be a valuable tool for a patient’s follow-up and personalized treatments and also when comparing singular interventions using a reference group. Clustering information and prototypes can be used in a subject’s examination as a means of classifying simple distance measurements of the patient’s features from the prototypes or clustering distributions for creating fuzzy rules or indexes. To the best of our knowledge, this is the first study demonstrating the potential prognostic and/or diagnostic utility of BIA in osteoporosis through clusterization. However, the small sample size can be considered a limitation of this study; nevertheless, we think that it is important to show how much clustering can represent an innovative method to predict or diagnose different pathologies, especially due to the fact that the analysis is a non-invasive and relatively inexpensive technique. We also want to add that the developed algorithm generated further results, and other studies are needed to interpret them and propose possible classification solutions.

It is important to clarify that it is not necessary to make an accurate diagnosis in osteoporosis, but it is necessary to find in advance the risk of developing this pathology and to induce people to change lifestyle and to reduce body weight. Indeed, osteoporosis is very often a “silent” disease, the clinical symptoms of which appear in the chronic stages of the disease; moreover, this condition can only be diagnosed through a combination of objective and sometimes subjective clinical tests. For this reason, having a predictive model of the risk of disease development through a non-invasive method such as BIA and its clusterization could have important implications.

## Figures and Tables

**Figure 1 diagnostics-13-02292-f001:**
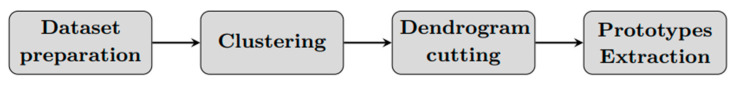
Clustering algorithm scheme.

**Figure 2 diagnostics-13-02292-f002:**
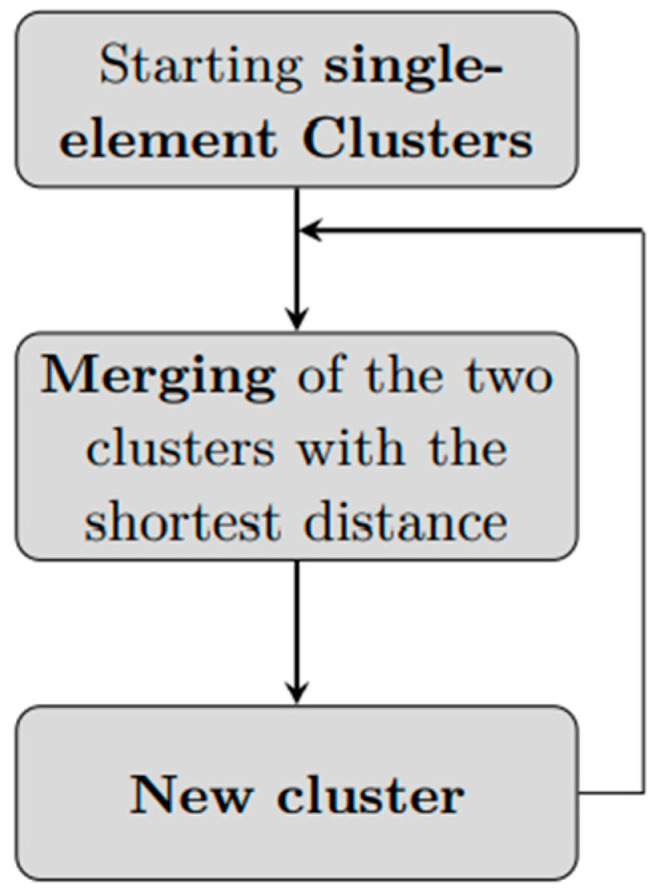
An agglomerative hierarchical clustering workflow. Starting from n clusters of cardinality 1, the two elements with the shortest distance are merged to create a new cluster of 2 elements. The next iteration will have *n* − 1 clusters of cardinality 1 and one cluster of cardinality 2, and, again, the two clusters with minimum distance will be merged. The process is repeated until reaching a single cluster with n elements.

**Figure 3 diagnostics-13-02292-f003:**
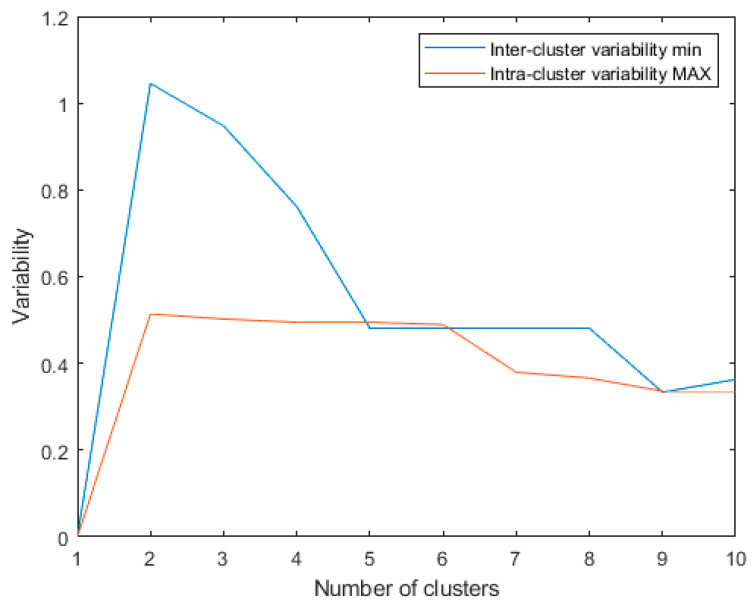
Minimum inter-cluster variability and maximum intra-cluster variability for the last 10 algorithm iterations using: metric: Euclidean; linkage: single.

**Figure 4 diagnostics-13-02292-f004:**
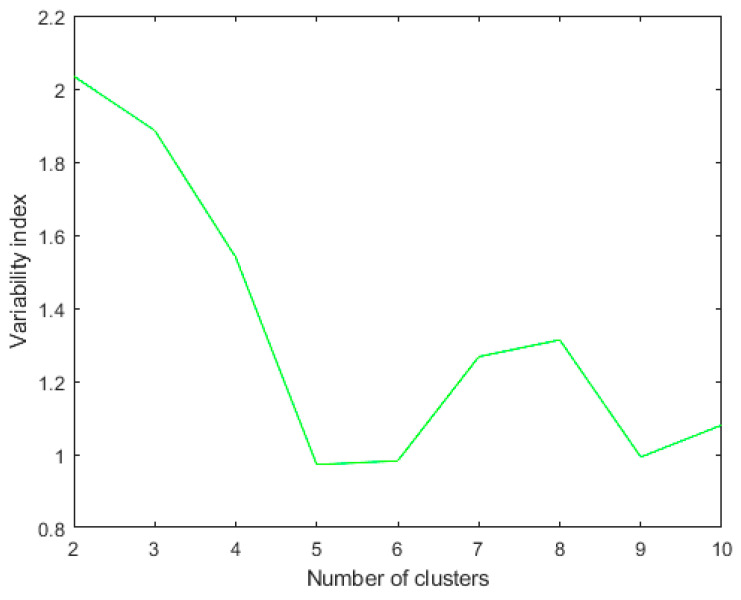
*R_k_* index for the last ten iterations using metric: Euclidean; linkage: single. The clustering algorithm is applied to a dataset of 46 patients. The best *R_k_* index in this case of two clusters.

**Figure 5 diagnostics-13-02292-f005:**
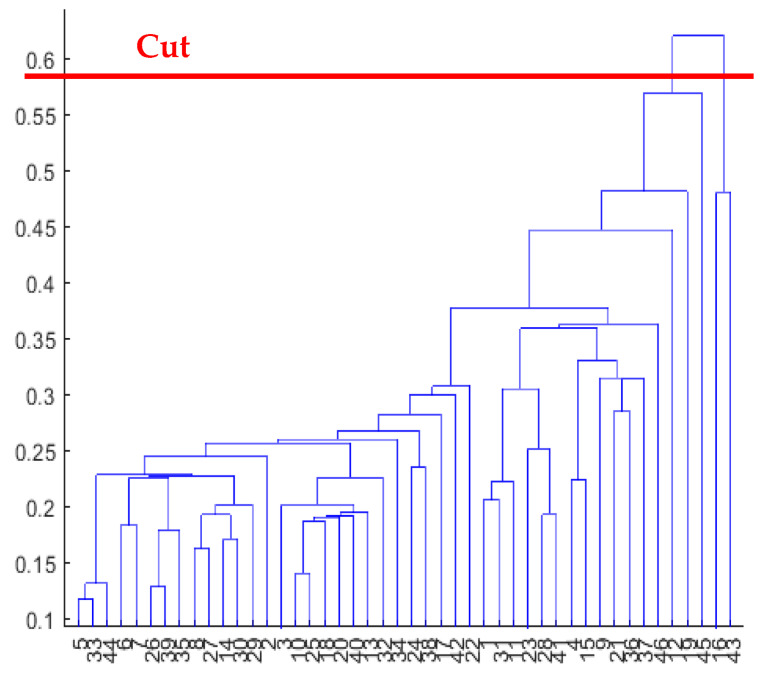
Dendrogram obtained using metric: Euclidean; linkage: single and cutting.

**Figure 6 diagnostics-13-02292-f006:**
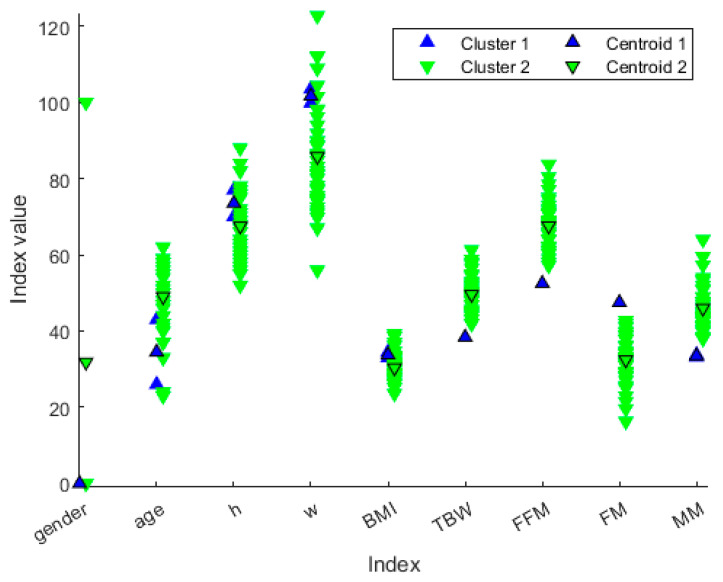
Distribution of patient values along the variables represented in different colors based on the cluster membership. The values with black marker edge color represent the cluster centroids. The gender is represented as 0 for female and 100 for male, whereas the height is represented in the form 1.x cm, so that, for example, *h* = 70 means a person or centroid 170 cm tall.

**Figure 7 diagnostics-13-02292-f007:**
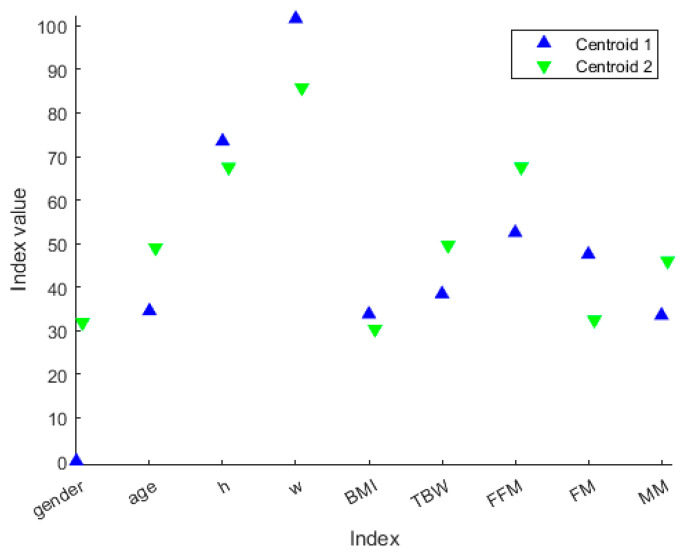
Centroid values along the variables represented in different colors based on the cluster membership. The values with black marker edge color represent the cluster centroids. The gender is represented as 0 for female and 100 for male, whereas the height is represented in the form 1.x cm, so that, for example, *h* = 70 means a person or centroid 170 cm tall.

**Figure 8 diagnostics-13-02292-f008:**
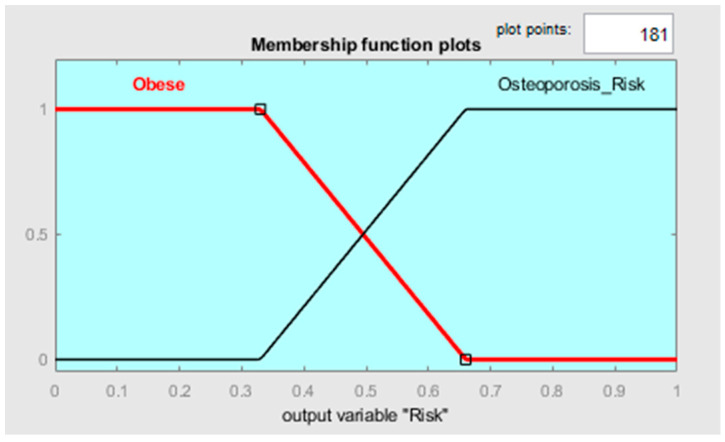
MF shapes along the output variable.

**Figure 9 diagnostics-13-02292-f009:**
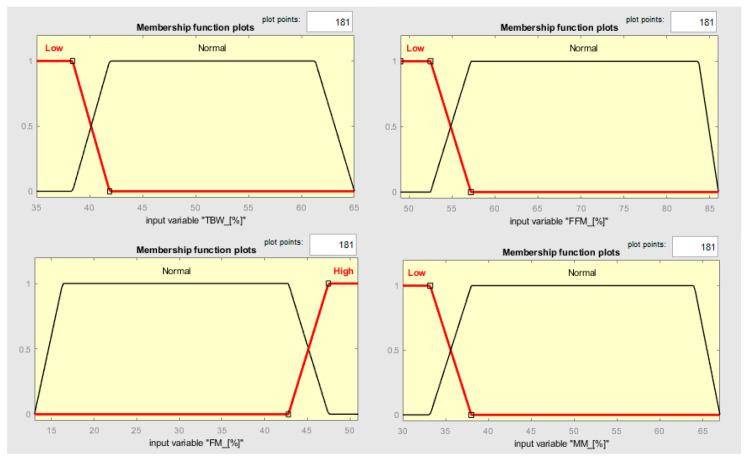
MF shapes along the input variables TBW, FFM, FM, and MM.

**Figure 10 diagnostics-13-02292-f010:**
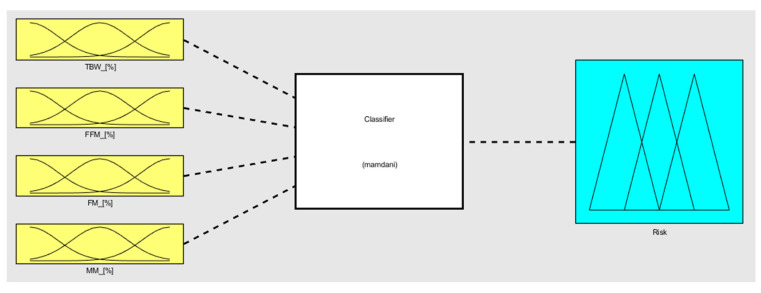
Fuzzy classifier in its wholeness.

**Table 1 diagnostics-13-02292-t001:** Metrics used to compute the distance between two generic elements *p_a_* = (*p_a_*_,1_, *p_a_*_,2_, …, *p_a_*_,*m*_) and *p_b_* = (*p_b_*_,1_, *p_b_*_,2_, …, *p_b_*_,*m*_).

Metric	Formula
Euclidean	${\left\|p_{a}-p_{b}\right\|}_{2}=\sqrt{\sum_{i=1}^{m}{\left(p_{a,i}-p_{b,i}\right)}^{2}}$
Squared Euclidean	${\left\|p_{a}-p_{b}\right\|}_{2}^{2}=\sum_{i=1}^{m}{\left(p_{a,i}-p_{b,i}\right)}^{2}$
Cityblock	${\left\|p_{a}-p_{b}\right\|}_{1}=\sum_{i=1}^{m}\left|p_{a,i}-p_{b,i}\right|$
Chebyshev	${\left\|p_{a}-p_{b}\right\|}_{\infty }={max}_{i}|p_{a,i}-p_{b,i}|$
Correlation	${\left\|p_{a}-p_{b}\right\|}_{Corr}=1-\frac{(p_{a}-{\bar{p}}_{a})(p_{b}-{\bar{p}}_{b}{)}^{\prime }}{\sqrt{(p_{a}-{\bar{p}}_{a})(p_{a}-{\bar{p}}_{a}{)}^{\prime }}\sqrt{(p_{b}-{\bar{p}}_{b})(p_{b}-{\bar{p}}_{b}{)}^{\prime }}}$

**Table 2 diagnostics-13-02292-t002:** Linkage criteria used to compute the distance between two generic clusters *A* and *B*. Here, *A* and *B* represent the cardinality (number of patients) of the clusters, while *C_A_* and *C_B_* are the clusters centroids.

Linkage Criterion	Formula
Single	$min\left\{d\left(p_{a},p_{b}\right)\;:\;p_{a}\in A,p_{b}\in B\right\}$
Complete	$max\left\{d\left(p_{a},p_{b}\right)\;:\;p_{a}\in A,p_{b}\in B\right\}$
Average	$\frac{1}{\left|A\right|\left|B\right|}\sum_{a\in A}\sum_{b\in B}d(p_{a},p_{b})$
Centroid	$C_{a}-C_{b}$

**Table 3 diagnostics-13-02292-t003:** Combinations of distance-linkage criterion tested. Each cell represents either the number of clusters corresponding to the higher *R_k_* or the type of clusterization results obtained. Methods providing the same clusterization are indicated with same cell colors.

			Metric		
Linkage Criterion	Euclidean	Squared Euclidean	Cityblock	Chebychev	Correlation
Single	2 Clusters (Clustering 1)	2 Clusters (Clustering 1)	2 Clusters (Clustering 1)	2 Clusters (Clustering 2)	2 Clusters (Clustering 2)
Complete	4 Clusters (Clustering 1)	4 Clusters (Clustering 1)	8 Clusters	4 Clusters (Clustering 2)	3 Clusters (Clustering 3)
Average	2 Clusters (Clustering 1)	2 Clusters (Clustering 1)	3 Clusters (Clustering 1)	4 Clusters (Clustering 3)	3 Clusters (Clustering 2)
Centroid	2 Clusters (Clustering 1)	2 Clusters (Clustering 1)	2 Clusters (Clustering 1)	4 Clusters (Clustering 4)	3 Clusters (Clustering 2)

**Table 4 diagnostics-13-02292-t004:** Principal anthropometric features of obese participants.

	Obese Subjects
M/F	14/32
Age	48.39 ± 9.01
Height (cm)	167.80 ± 8.45
Weight (kg)	86.47 ± 13.57
BMI (kg/m^2^)	30.48 ± 3.48
TBW (%)	49 ± 5.20
FFM (%)	67 ± 6.90
FM (%)	33 ± 6.90
MM (%)	45 ± 6.43

**Table 5 diagnostics-13-02292-t005:** Fuzzy rules representing the connections among input and output variables.

	Input Variables					Output Variable
	TBW	FFM	FM	MM		Risk
if	Low	Low	High	Low	then	Osteoporosis
if	Normal	Normal	Normal	Normal	then	Low

**Table 6 diagnostics-13-02292-t006:** Experiment data used to determine the coefficients of Equation (8).

	TBW	FFM	FM	MM	Risk
Centroid	49.58	67.57	32.43	45.99	0
Closest values No-Risk on Risk	42	57.2	42.8	38	0.2
Centroid	38.4	52.5	47.5	33.5	0.9
3 percentage points over Risk	35.4	49.5	50.5	30.5	1

## Data Availability

The Data are available at the corresponding authors.

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
