# Peer review of "The Role of BIA Analysis in Osteoporosis Risk Development: Hierarchical Clustering Approach"

_diagnostics, 2023, doi:10.3390/diagnostics13132292_

Round 1
Reviewer 1 Report (Previous Reviewer 1)
Thank you for considering the comments. In the sent version, I suggest moving the group characteristics to Results and providing data in the table, not in the charts that relate to individual characteristics. In my opinion, it is worth analyzing and writing on what basis "obese" was established. Added BMI histograms show values for overweight diagnosis, but FM values without gender, suggest misdiagnosis.
Author Response
We thank the reviewer for these suggestions.
In attached our response.
Best regards

Reviewer 2 Report (New Reviewer)
The examination of alternative methods to manage osteoporosis is important and the authors describe an initial study of using BIA and clustering techniques.
1. Overall, the manuscript requires extensive editing for the proper use of English. Some errors noted:
Line 16: among of these lifestyle and obesity.
37: wrong lifestyle
41: in literature were
59: is the result
67: Because that
93: in literature
113: Instead
124; 185; 324-5: The sentences need restructuring.
186: accurately is not the correct word.
215, 416: dosage is not correct. You mean blood level.
Many typographical errors: Lines 510, 511, 516, 517, 519, 551, 553, 592.
565: bloodless is not the appropriate word.
324: include citations for "Given the strong correlation between body composition, obesity, osteoporosis, and vitamin D, supported by many scientific research"
2. The introduction should describe the common use of DXA for measuring BMD and monitoring bone health.
3. Figures are numbered incorrectly
4. Figures 1-6 should be located in Results.
5. Line 415: Cite your prior research study.
6: Line 324: include citations for "Given the strong correlation between body composition, obesity, osteoporosis, and vitamin D, supported by many scientific research"
7. Provide a full description of the lab protocol for participants. Also describe how they were recruited.
I defer examination of the clustering analysis to other reviewers who are experienced in this area.
Extensive editing is required.
Author Response
We thank the reviewer for these suggestions.
In attached our response.
Best regards

Round 2
Reviewer 2 Report (New Reviewer)
The authors have addressed most of my concerns. English errors persist and editing is needed. For example:
Line 650: These evidence further reinforce
Line 660: BIA analysis it is possible evaluate FFM, this parameter is positive associated with BMC. This association has important for endocrine activity of muscle.
684: knoweledg
English errors persist and editing is needed. For example:
Line 650: These evidence further reinforce
Line 660: BIA analysis it is possible evaluate FFM, this parameter is positive associated with BMC. This association has important for endocrine activity of muscle.
684: knoweledg
Author Response
Dear reviewer,
we corrected everything that was requested.
Hoping to have satisfied your requests.
Thank you
Best regards
This manuscript is a resubmission of an earlier submission. The following is a list of the peer review reports and author responses from that submission.
Round 1
Reviewer 1 Report
This is interesting manuscript with an interesting approach to the development of data from BIA, but I have a few comments:
Introduction: I don't see a need to write about vitamin D if it was not study; the aim requires simplification in accordance with the study.
Material and Methods: lack of detailed data on the research and measurement methodology used and the approval of the ethics committee. What does "obese people" mean? what criteria were used?, please add cut-off points and reference values. What type of BIA device was used? - add name and symbol. Under what conditions was the BIA procedure performed? Whether a repeated measurement has been performed? Were and how the patients adequately prepared? What was their hydration status?
Results: where I find it? There is lack of group characteristics, and also BIA results for clusters
Lines 331--3; "Given the interesting results, and above all the FM equal to 50 % of the total weight, we confirmed these data also with the DEXA and with the dosage of vitamin D, which was found to be on average 18 ng/mL." - there is no mention of this before, and it would be worth referring the obtained results to these data and presenting them in the results
References - needs updating
Figure 6 is unreadable.
Reviewer 2 Report
First of all, I would like to thank the authors for the presented results of their investigation related to BIA analysis and osteoporosis risk development. Also, I would like to thank the editor for the opportunity to review this manuscript.
The manuscript entitled “The role of BIA Analysis in Osteoporosis Risk Development: Hierarchical clustering approach.” presents a hierarchical clustering method that, using BIA parameters, can organize patients with similar characteristics. In my opinion, the authors present an interesting topic that falls within the aims and scope of the Diagnostics journal. Furthermore, since osteoporosis has been studied from various aspects, using BIA parameters for putting new insight into the topic is always welcome.
Although the manuscript is well structured, my major concern is related to wholeness. Namely, the research was cut short when the main findings were to be presented. A scientific presentation of this type must offer more specific indicators than simply answering whether it is possible to use some statistical method. Unfortunately, putting the focus of the paper this way on the statistical method could not be enough to recommend the publication.